# The Creation of Fundamentally New Products as a Factor of Organizations' Sustainable Economic Development

**Alexander Chursin, Zhanna Chupina \*, Anna Ostrovskaya and Andrew Boginsky**

Higher School of Industrial Policy and Entrepreneurship, Peoples' Friendship University of Russia (RUDN University), 6 Miklukho-Maklaya Street, Moscow 117198, Russia; cursinaleksandr76@gmail.com (A.C.); ostrovskaya-aa@rudn.ru (A.O.); boginskiy-ai@rudn.ru (A.B.)
\* Correspondence: vse.1@mail.ru; Tel.: +7-977-5525618

**Abstract:** This study analyzed the sustainable economic development of some organizations for the period of 2020–2022. The authors found that organizations' sustainable development is influenced by technological superiority based on the creation and production of radical new products that can form new markets or be dominant in existing ones. This study examined effective management based on the application of digital technology and artificial intelligence for the creation and production of radically new products, which creates conditions for the advanced sustainable economic development of the organization. The main drivers of these processes are technological platforms, the formation of which requires significant amounts of different types of resources. To solve the issue of investment in the creation of radically new products in conditions of limited resources, the authors researched and developed tools for the effective use of investment in the creation of radically new products in order to ensure the formation of organizations of advanced, sustainable economic development. In the development of methodological tools for managing the creation and development of radically new products, a conceptual mathematical model for assessing the criteria of economic efficiency of projects is proposed. Thus, the proposed tools for managing the creation of radically new products and advanced sustainable economic development of organizations form the basis for technological superiority and sustainable economic development in modern economic conditions.

**Keywords:** sustainable development; technological superiority; technological platform; radical new products; risk; unique products; innovation development; investment project; risk management; uncertainty; economic efficiency

## 1. Introduction

The contemporary global economic situation is dominated by the USA, China, and the European Union, which take a leading role in international markets for the export of equipment, vehicles, and other goods; as such, these economies exhibit GDP growth. However, there are still certain countries which dominate market niches through the sale of raw materials, food products, and resources, such as Russia. In this study, we consider the factors that contribute to the growth of GDP, those countries that exhibit a positive tendency to increase GDP, and those countries that demonstrate sustainable economic development.

This study formulates proposals on how to increase GDP and thereby accelerate national processes of sustainable economic development. This study offers a toolkit for creating radically new products and proposes directions for the accelerated development of certain organizations. In this case, the authors proposed a methodology presented as a conceptual mathematical model for evaluating the economic efficiency criteria of projects. The research focuses on an example project of implementing a technological platform, the implementation of which will solve the problems associated with arranging interactions between all participants in a unified scientific and production system, which has the main goal of creating value for the consumer in the form of radically new products.

The purpose of this study was to determine the theoretical and practical foundations of the creation of radically new products and the implementation of the processes of advanced enterprising development, taking into account the application of mathematical modeling of resource provision.

The research considered two main questions: (1) Is it possible to effectively manage the creation and production of radically new products through the application of digital technology and artificial intelligence? and (2) How can we increase GDP and thereby accelerate processes toward sustainable economic development? The following section presents a literature review and the development of the hypotheses; Section 3 elucidates the methodology; and Section 4 presents the case study results and the discussion.

## 2. Literature Review and Development of the Hypothesis

In the contemporary economic climate, GDP is the key indicator used to measure national socio-economic development and reflect sustainable economic development [1].

The issues of sustainable economic development have been addressed by a number of researchers. According to [2], the concept of sustainable socio-economic development is associated with satisfying the material and spiritual needs of the population. In accordance with this, sustainable development can be perceived as a socio-economic concept and is defined as a system of economic relations that ensure the continuous maintenance of sustainability or economic growth under conditions of optimal proportionality at minimum cost and environmental safety, contributing to more complete satisfaction of the material and spiritual needs of the population of the country and its regions.

According to [3], the essence of sustainable development is to ensure high indicators of social, economic, and environmental conditions of a country or region over a long period of time.

For example, [4] notes that "sustainable development is a new type of social development in which the achievement of a stable socio-economic state in a country or region, constituting the goal of development, at the same time should create reliable prerequisites for sustainable development in the long term future".

Robert Costanza et al. (1992) [5] in "Natural Capital and Sustainable Development" defined the essence of the concept as "development without growth", i.e., "socially sustainable development in which gross economic growth should not exceed the carrying capacity of life support systems". Granberg, A. G. et al. (2002) [6] provide the following definition of the term: sustainable development is "stable balanced socio-economic development that does not destroy the natural environment and ensures the continuous progress of society". Zhu, J. et al. (2023) [7] explored sustainable economic development and energy efficiency in China, determining that these concepts are influenced by the sharing economy, which is characterized by the fusion of information, marketing, and technology to create a new paradigm in which consumers value access over ownership, enabling them to make better use of resources. It represents a modern business strategy that can pave the way for energy efficiency and sustainable economic development. Pullinger, M. (2014) [8] indicated that sustainable economic development can be influenced by reducing per capita consumption, especially among high-income groups, which is often considered necessary to reduce the impact of the global economy on the environment. Pullinger described the conditions under which a reduction in working time can benefit the environment and well-being, then provided examples of innovative policies of voluntary reductions in working time in the Netherlands and Belgium. Kamal, M. M. et al. (2022) [9] suggested that SME electrical and electronic equipment manufacturers cannot become more sustainable due to their high e-waste rates. On the other hand, consumers play an important role in SME manufacturers meeting sustainability goals because they are responsible for taking their e-waste back to SMEs. This study explores the type of information that influences consumers' intentions to immediately return their e-waste back to SME producers, which affects companies' sustainable production. Hafezali Iqbal Hussain et al. (2023) [10], in their paper entitled "Does income inequality influence the role of a sharing economy in

promoting sustainable economic growth? Fresh evidence from emerging markets", reflected that the impact of the sharing economy is becoming increasingly prominent in the promotion of sustainable economic development. This study examined this relationship in the context of emerging markets, analyzing the impact of income inequality on limiting the expected benefits of asset or service-sharing activities. The results showed that developing effective and efficient platforms would enable developing countries to take advantage of the resource-sharing economy.

Thus, we can conclude that these researchers have associated sustainable economic development with GDP growth, and some factors of this growth include the production and sale of knowledge-intensive and high-tech competitive products. Exploring this position, we present the relevant statistical data in Table 1.

**Table 1.** GDP (PPP) of the world's top ten largest economies for 2020–2022.

| No. | Country | 2020 (Billion USD) | 2021 (Billion USD) | 2022 (Billion USD) | Growth Rate 2020/2021 (%) | Growth Rate 2021/2022 (%) |
|---|---|---|---|---|---|---|
| 1 | China | 24,168.03 | 27,206.27 | 30,177.93 | 112.6 | 110.9 |
| 2 | USA | 20,893.75 | 22,997.50 | 25,346.81 | 110 | 110.2 |
| 3 | India | 9005.11 | 10,218.62 | 11,745.26 | 113.5 | 114.9 |
| 4 | Japan | 5304.97 | 5615.00 | 6110.08 | 105.8 | 108.8 |
| 5 | Germany | 4536.52 | 4856.77 | 5269.96 | 107 | 108.5 |
| 6 | Russia | 4117.75 | 4490.46 | 4365.44 | 109 | 97.2 |
| 7 | Indonesia | 3302.09 | 3566.28 | 3995.06 | 108 | 112 |
| 8 | United Kingdom | 3040.72 | 3402.76 | 3751.85 | 111.9 | 110.2 |
| 9 | Brazil | 3153.14 | 3435.90 | 3680.94 | 108.9 | 107.1 |
| 10 | France | 3016.96 | 3361.63 | 3677.58 | 111.42 | 109.3 |

Source: authors' calculations.

The international data in Table 1 present positive dynamics of GDP over three years. The GDP growth rate of the presented countries ranged from 7% to 14.9% (India) in 2022 compared with 2021. Let us take a closer look at the four leading countries from this list: China, the USA, India, and Japan.

In total, 90% of manufacturing products were exported from the USA; 60% of automobiles and transport equipment were exported from Japan; 50% of industrial products were exported from India; and 30% of mineral resources, including fuel, were exported from China, as well as 59% of total exports of mechanical and electrical products. These countries send most of their exports to the global market of science-intensive and high-tech products. Studying the export structures of four more countries from the list, namely Russia, Brazil, Indonesia, and France, reveals that the extraction and supply of energy, minerals, agricultural products, etc., prevails.

Altogether, four Asian countries (China, Japan, the Republic of Korea, and Taiwan) account for 32% of the global output of high-tech goods and services, which corresponds to the same contribution of the USA and significantly exceeds the share of all European countries. The share of Asia as a whole in global exports of products of this category already exceeds 50%, as many Asian countries are mainly focused on exports of high-tech products. The USA dominates the manufacture of computers, electronics, and optical products (30%); publishing, including the publication of computer programs (25%); and pharmaceuticals (17%). The USA still leads the world in almost all high-tech industries (with the exception of computers, electronics, and optical products, where China leads).

Comparing the eight countries characterized by GDP growth, the rates of GDP growth and labor productivity growth are higher in the first group of countries because these countries produce high-tech products. Accordingly, in order to manufacture such products, advanced means of production and a sizeable production capacity are necessary. This suggests that these countries invest in research and development and educational processes,

which enables the creation of equipment and competitive products that occupy major niches in the global market or form new markets: the smartphone market, the computer market, the electric car market, etc.

Thus, technological superiority is determined by the number of advanced critical technologies and their development, as well as innovation in high-tech and competitive products.

Technological superiority represents an opportunity to dominate the market of science-intensive products with high added value, to export products to many countries around the world [11]. As a result, the export of such products reaches approximately 50–60% in the composition of gross domestic product (GDP).

Previously, the authors studied the methods, techniques, and means of creating products, dominating a market, and capturing a market, and provide a definition of radically new products.

Radically new products have technical characteristics superior to those of their competitors; consumer properties focus on satisfying both current and prospective needs at a high level, ensuring that the cost of purchase and ownership are accessible to the consumer, and a capability of dominating or creating new markets.

The creation of radically new products, with unique properties, is based on the processes of building innovation capacity of the organization and the formation of competencies, which, in turn, determine the creation of competitive advantages of products.

To achieve the sustainable economic development of an organization, it is necessary to create technical and economic conditions that enable the development and production of radically new products and continually update the processes, ensuring the long-term dominance of these products in sales markets [5,12,13].

*Hypothesis*

Technological superiority is impossible without the creation of a unified technological platform, which influences the development and production of radically new products and processes ahead of the development of the organization.

## 3. Research Methodology

In turn, the competitiveness of an organization is determined not only by radically new products (highly competitive), but also by a number of other factors influencing the capabilities of a system of putting new products into production, particularly the integral indicators of management efficiency of the processes of devising the technical and economic appearance of the product ($K_1$), as well as the processes of designing and constructing the product ($K_2$), preparing the production and manufacturing of the products ($K_3$), and the life cycle ($K_4$).

With the known indicators of competitiveness of the entire set of products, $\widetilde{IQ}$, and estimates of the integral indicators that determine the functional capabilities of the technical and cyber-economic system of the organization, $K_1$, $K_2$, $K_3$, and $K_4$, the integral indicator of competitiveness of an organization can be written in the form:

$$IQO = w_0\widetilde{IQ} + w_1K_1 + w_2K_2 + w_3K_3 + w_4K_4 + w_5IO,$$

where $w_i$ is the weight coefficient of factors of competitiveness of the organization; thus, $\sum_{i=0}^{5} w_i = 1$.

We consider an organization capable of achieving global competitive superiority in the market that uses a decision support system for the production of new products, whose integral competitiveness index is $IQO \geq 1$.

Evaluating the dynamics in relation to a period of $M$ years, the formula for assessing the competitiveness of a technical and cybereconomic system of the organization can take the form:

$$IQS = \frac{\Delta IQO_t}{\frac{1}{M}\sum_{i=0}^{M-1} \Delta IQO_{t-i}},$$

where $\Delta IQO_\tau = IQO_\tau - IQO_{\tau-1}$.

The successful implementation of a project for creating a highly competitive product, which could displace existing products or create a new market, directly depends on the availability of key competencies and the level of scientific and technological potential, due to which unique technical, economic, and consumer characteristics of new products can be created. Thus, the foundation of advanced development is the competitiveness already instilled within a company.

Radically new products require appropriate creation and provision of the means of production, and these means of production are also unique products; only with the help of highly technical means of production is it possible to create such products [14–16].

Thus, the creation and production of radically new products in advancing the development of an organization creates conditions in which innovative technologies are developed and implemented in practice, providing an output of radically new products and, as stated above, creating conditions for the advanced development of an organization. This ensures the sale of large volumes of products at a price determined by the sales market, taking into account the high knowledge content and high consumer demand, as they dominate the market. This promotes sustainable economic development of the organization and ensures the possibility of directing this profit toward the creation of new competences and progressive technologies, which can surpass the already-achieved technological level of the organization's development [17–19].

To solve the problems of technological development, numerous studies are currently ongoing in the field of creating technological platforms.

A technological platform is understood as a communication platform for interaction between businesses, science, consumers, and the state on the issues of modernization and scientific and technological development in certain technological areas [20,21].

In this study, we consider the possibility of fostering approaches to promote the development and application of technological platforms.

The creation and development of technological platforms at the industry level addresses problems associated with the organization of interaction between all industry participants within a unified scientific and production system with the main goal of creating value for the consumer in the form of a radically new product on the basis of cross-platform interaction of business structures of the industry. The creation of new technological platforms makes it possible to conceive unique products that can form new markets or be dominant in existing markets as a result of applying radical competences.

The implementation of these technological platforms and the innovative solutions that are created and applied within these platforms requires a large amount of investment; the amount of investment depends on the multidimensional nature of the technological platform, the complexity of the radically new products being developed, the technical and technological level of the organization, etc. [22–25].

Based on the statement of the problem, the creation of a sectoral technological platform, we can approach the issue that this is a major investment project aimed at creating radically new products and ensuring the processes of advanced development of the organization of both large resources of the state and business.

Determining estimates of the initial financial and economic parameters of this investment project is an important approach in a situation of non-stochastic uncertainty. One of the main approaches to the formalization of information in a situation of non-stochastic uncertainty is the construction of subjective probabilistic and fuzzy-multiple estimates of the initial parameters of the problem situation under study.

Subjective probability involves two basic methods of formalization: qualitative and quantitative. Depending on the nature of the questions posed to the expert(s), three groups of methods for obtaining subjective probabilities can be distinguished.

The first group consists of methods in which the expert, in answering the questions posed, directly operates with the probabilities of the events being analyzed. This group is the most extensive in its composition. The methods of this group include the direct esti-

mation method, the variable interval method, the fixed interval method, the ratio method, the method of pairwise (paired) comparisons, the method for estimating distribution parameters, methods based on P. Fishburn formulas, the graphical method, etc.

The methods of the second group consist of obtaining event probabilities based on experts' decisions in some hypothetical choice situations, in which the result of the alternatives under study is the realization of a random variable. Purely formally, this approach does not compare the probabilities of events as such, but the expected utility of alternatives. An indicative representative of the methods of this group is the so-called method of equal baskets. The application of these methods, due to their specificity, requires more caution than the methods of the first group.

The third group includes methods that combine aspects of the previous two approaches, i.e., the expert has to operate with both probabilities and utilities.

It is reasonable to describe some of these methods in more detail. Let us begin with those that determine probabilities for a finite set of incompatible events.

Let $A_i$, $i = \overline{1,n}$ be the complete list of incompatible events to be analyzed; $p_i$, $i = \overline{1,n}$ are the probabilities of realization of the events in the specified list, with $P(A_i) = p_i$, $i = \overline{1,n}$, where $P(\ldots)$ is the probability of the corresponding event.

In direct estimations, the expert must sequentially determine the probabilities, $p_i$, $i = \overline{1,n}$, of all events, $A_i$, $i = \overline{1,n}$. The described method implies different options. According to one of the possible options, the most probable event is first selected from the list of events under study, after which its probability is evaluated. Then, this event is removed from the initial list, and the procedure is applied to the remaining events. This continues until all events have been considered.

In the relationship method, the expert first chooses the most probable event, which is assigned an unknown probability, $p$. Then, the expert has to successively estimate the ratio of probabilities of all other events to this probability: $p_i/p$. Based on the values of these ratios and the normalization condition, $\sum_{i=1}^{n} p_i = 1$, equations can be derived to find the probability of $p$; then, all other probabilities are calculated.

The scope of the methods presented above is the situation of a finite set of incompatible events. If, however, the random variable under analysis is assumed to be continuous, then subjective probabilities can be found based on the variable interval method, the fixed interval method, and other methods.

Let us now consider the issue of assessing the initial financial and economic parameters of the investment project with the tools of the theory of fuzzy sets [24].

Within the framework of the fuzzy-multiple methodology, a fairly well-developed methodological apparatus for the construction of the membership functions of fuzzy sets has been formed. The main differences between the methods are related to the fact that in the direct or indirect (indirect) method, the membership functions are found, and one or more experts carry out their evaluation. The direct method consists of the assignment of membership functions by either directly assigning degrees of membership to individual values of the studied characteristic (object), or by means of an analytical expression.

Let $\widetilde{A}$ be some fuzzy set, the carrier of which consists of a finite number of real numbers whose membership function values are unknown:

$$\widetilde{A} = \left\{ (x_i, \mu_{\widetilde{A}}(x_i)) \,|\, x_i \in \Re, \mu_{\widetilde{A}}(x_i) \in [0,1], i = \overline{1,n} \right\}, \tag{1}$$

where $x_i - i$-n is the element of fuzzy set $\widetilde{A}$; $\mu_{\widetilde{A}}(x_i)$ is the degree to which the element (number) belongs to $x_i$ of fuzzy set $\widetilde{A}$; and $\Re$ is scale of variation for nondeterministic (probabilistic, interval, and fuzzy) estimations of economic indicators.

The sequence of stages (steps) in finding the membership function based on the method of pairwise comparisons is as follows.

First, the elements selected for analysis at the level of verbal evaluations are compared in pairs with each other to determine the degree of their importance (significance and priority) in relation to the fuzzy set under study, as well as their suitability. Table 2 shows

the evaluation concepts, as well as their quantitative interpretations, which should be used in this case.

**Table 2.** Estimated concepts and their quantitative interpretation within the method of pairwise comparisons.

| No. | Ratio of Importance | Qualitative Assessment |
|-----|---------------------|------------------------|
| 1 | 0 | Incomparability |
| 2 | 1 | Equally important |
| 3 | 3 | Somewhat more importantly |
| 4 | 5 | More importantly |
| 5 | 7 | More importantly |
| 6 | 9 | Absolutely more important |
| 7 | 2, 4, 6, 8 | Interim assessments |
| 8 | 1, 1/2, 1/3, 1/4, 1/5, 1/6, 1/7, 1/8, 1/9 | Inverse values of the corresponding of the corresponding non-zero estimates |

Sources: authors' calculations.

After translating qualitative assessments into numbers, the comparison results are displayed in a so-called matrix of pairwise (paired) comparisons:

$$W = \begin{pmatrix} w_{11} & w_{12} & \dots & w_{1n} \\ w_{21} & w_{22} & \dots & w_{2n} \\ \dots & \dots & \dots & \dots \\ w_{n1} & w_{n2} & \dots & w_{nn} \end{pmatrix}, \tag{2}$$

where $w_{ij}, i \in \{1, \dots, n\}, j \in \{1, \dots, n\}$ is a number which is the result of comparing the degree of membership of the fuzzy set element $\widetilde{A}$ with element $x_i$ with the degree of membership of the element, $x_j$.

At the last stage, we search for the eigenvector of the matrix $W$, which corresponds to its largest eigenvalue and satisfies the normalization requirement. Consequently, the equation for its determination has the form:

$$Wr = \lambda_{\max}r, \tag{3}$$

where $r = (r_1, r_2, \dots, r_n)$ is the eigenvector of matrix $W$, for which it is assumed that $\sum_{i=1}^{n} r_i = 1$; $\lambda_{\max}$ is the largest eigenvalue of matrix $W$.

The identified values, constituting the eigenvector $r$, are taken as estimates of the degree of membership of the elements $x_i, i = \overline{1, n}$ of fuzzy set $\widetilde{A}$:

$$\mu_{\widetilde{A}'}(x_i) = r_i, i = \overline{1, n}. \tag{4}$$

Suppose that for the fuzzy set $\widetilde{A}$, which is described by the matrix of pairwise comparisons $W$, an additional restriction is set: the requirement of normality (i.e., the maximum value of its membership function must be equal to 1).

In this case, taking into account the previous estimate of the degrees of belonging of the given set can be determined on the basis of the following relationship:

$$\mu_{\widetilde{A}}(x_i) = \frac{r_i}{r_{\max}}, i = \overline{1, n}, \tag{5}$$

where

$$r_{\max} = \max\{r_i | i = \overline{1, n}\}. \tag{6}$$

Both the method of pairwise comparisons and other methods developed within the framework of the general theory of affiliation functions construction can be successfully used in investment analysis. At the same time, the fuzzy-multiple modeling problem of

initial financial and economic parameters of investment projects has its own peculiarities; thus, the appeal of the general approaches should be supplemented with the development and use of special methods.

The above considerations affirm the results of the review of various scientific papers devoted to the economic analysis of real investments using the theory of fuzzy sets, which, on the one hand, did not identify any systematic methodology of the fuzzy-multiple evaluations of initial investment project data, but on the other hand, identified the main current trends on this issue.

The starting point in studies performed by different authors is an arsenal of methods for constructing the membership functions of fuzzy sets, which provide a wide range of possible applications, including those that can be used in tasks of investment analysis.

Secondly, separate attention can be paid to methodological design and development, for which the problem of economic evaluation of real investments is considered one of the priorities. One example of such a method is the fuzzy-multiple modeling of initial financial and economic parameters of investment projects based on limited statistical data.

Third, an approach to the evaluation of investment projects based on the use of fuzzy mathematical models. This approach involves uncertainty in the initial data of the project and the calculation of project success probabilities using fuzzy logic and fuzzy statistics methods.

Fourth, risk analysis methods using fuzzy sets can be used to evaluate investment projects. Such analysis considers the uncertainties and risks associated with investment, making more appropriate decisions in the selection of investment projects.

The general conclusion from the literature review is that fuzzy set modeling methods can be successfully applied in analyses of investment projects. However, it is necessary to consider the specifics of a task and develop special methods for solving specific investment problems.

For example, if it is predicted that the amount of income from an investment in a project will be approximately 2800 c.u. but will definitely be in the range of 2000 to 3500 c.u., then such information should be represented as a triangular number. If, however, the forecast states that the amount of income could be between 1500 and 6000 c.u. but most likely between 2500 and 4000 c.u., then it is appropriate to model this information with a trapezoidal number.

Triangular and trapezoidal numbers can be written as triples and quadruples, respectively:

$$\widetilde{A} = (a_{\min}, a_{\mod}, a_{\max}), \widetilde{B} = \left(b_{\min}, \underline{b}_{\mod}, \overline{b}_{\mod}, b_{\max}\right),$$

where $a_{\min}, a_{\max}$ are the lower and upper limits of the closing of triangular number carrier $\widetilde{A}$, respectively; $a_{\mod}$ is the modal value (moda) of triangular number $\widetilde{A}$ (for which the degree of affiliation acquires the greatest value); $b_{\min}, b_{\max}$ are the lower and upper limits of the trapezoidal number carrier closure, $\widetilde{B}$, respectively; and $\underline{b}_{\mod}, \overline{b}_{\mod}$ are the lower and upper limits of the interval within the trapezoidal number, $\widetilde{B}$, respectively, for which the degree of affiliation acquires the greatest value (i.e., the lower and upper modes of trapezoidal number $\widetilde{B}$).

Notably, in general cases, the exact upper bound of a membership function of a fuzzy number can take any value in the interval [0, 1]. However, analyses of investment projects are usually limited to normal fuzzy numbers, for which the upper bound tends to be equal to one. In the subsequent discussion, we proceed from the assumption of normality of fuzzy numbers.

Additionally, for the needs of this study, it is appropriate to define notions of the carrier and the kernel of fuzzy sets.

The carrier of a fuzzy set is the usual set which contains those, and only those, elements of the universal set (universum) for which the values of the membership function of the fuzzy set are non-zero.

The kernel of a fuzzy set is an ordinary subset of its elements for which the values of its membership function are equal to one. In analytical form, the accessory functions of triangular and trapezoidal numbers are defined by the following relationships:

$$
\mu_{\widetilde{A}}(x; a_{\min}, a_{\mod}, a_{\max}) = \begin{cases} 0, x \le a_{\min} \\ \frac{x - a_{\min}}{a_{\mod} - a_{\min}}, a_{\min} < x < a_{\mod} \\ 1, x = a_{\mod} \\ \frac{a_{\max} - x}{a_{\max} - a_{\mod}}, a_{\mod} < x < a_{\max} \\ 0, x \ge a_{\max} \end{cases}, \tag{7}
$$

$$
\mu_{\widetilde{B}}(x; b_{\min}, \underline{b}_{\mod}, \overline{b}_{\mod}, b_{\max}) = \begin{cases} 0, x \le b_{\min} \\ \frac{x - b_{\min}}{\underline{b}_{\mod} - b_{\min}}, b_{\min} < x < \underline{b}_{\mod} \\ 1, \underline{b}_{\mod} \le x \le \overline{b}_{\mod} \\ \frac{b_{\max} - x}{b_{\max} - \overline{b}_{\mod}}, \overline{b}_{\mod} < x < b_{\max} \\ 0, x \ge b_{\max} \end{cases}. \tag{8}
$$

In terms of generalized theoretical constructs of fuzzy mathematics, both triangular and trapezoidal numbers are special cases of the so-called fuzzy numbers $(L - R)$. A fuzzy number $(L - R)$-type object is a fuzzy number whose membership function can be represented as a composite of two functions of a real variable, $L : \Re \to [0, 1]$ and $R : \Re \to [0, 1]$, which are non-growing on the set of non-negative numbers and for which the following additional conditions are satisfied: $L(-x) = L(x), R(-x) = R(x); L(0) = R(0) = 1$. This distinguishes unimodal and tolerant fuzzy numbers as $(L - R)$ types. Specifically, the membership function of a unimodal fuzzy number of $(L - R)$ type has the form:

$$
\mu_{LR}^{U}(x) = \begin{cases} L\left(\frac{a_{\mod} - x}{\alpha}\right), x \le a_{\mod} \\ R\left(\frac{x - a_{\mod}}{\beta}\right), x \ge a_{\mod} \end{cases}, \tag{9}
$$

where $a_{\mod}$ is the modal value of a unimodal fuzzy number of $(L - R)$ type; $\alpha, \beta$ are the left and right fuzzy coefficients of the unimodal fuzzy number $(L - R)$ types, respectively, $\alpha > 0, \beta > 0$.

A significant property $(L - R)$ of fuzzy number representations that makes them an attractive tool for practical applications is the possibility to perform arithmetic operations on them through their parameters.

Another important special case of $(L - R)$-type fuzzy numbers is fuzzy numbers based on a Gaussian membership function, which is defined by the formula:

$$
\mu_{G}(x) = \exp\left[-\frac{(x - b)^2}{2a^2}\right], \tag{10}
$$

where $b$ is the modal value for membership function $\mu_{G}(x)$; $a$ is a parameter that displays a measure of the concentration (or scatter) of the fuzzy number values described by the membership function $\mu_{G}(x)$, with respect to the mode.

For example, asymmetric unimodal $(L - R)$ fuzzy numbers based on the Gaussian identity function have the form:

$$
\mu_{LR}^{UG}(x) = \begin{cases} \exp\left[-\frac{(x - b)^2}{2a_1^2}\right], x \le b \\ \exp\left[-\frac{(x - b)^2}{2a_2^2}\right], x \ge b \end{cases}, \tag{11}
$$

This assumes that $a_1 \ne a_2$.

In many situations, including those related to investment analysis problems, it is inconvenient to use the classical Gaussian membership function because of the unbounded

nature of its support. This disadvantage is derived by the Gaussian function with a limited carrier, which serves as a starting theoretical construction for the construction of a corresponding family of fuzzy numbers. The formula for this function is as follows:

$$\mu_{G^m}(x) = \begin{cases} 0, x \leq \lambda_1 \\ \exp\left[\frac{4(\lambda_2-x)(x-\lambda_1)-(\lambda_2-\lambda_1)^2}{4(\lambda_2-x)(x-\lambda_1)}\right], \lambda_1 < x < \lambda_2, \\ 0, x \geq \lambda_2 \end{cases} \tag{12}$$

where $\lambda_1$ and $\lambda_2$ are the lower and upper bounds of the closure of the membership function carrier, $\mu_{G^m}(x)$, respectively.

Similar to the case of the usual Gaussian function, on the basis of its considered modification, various derivative variants of fuzzy numbers with the necessary characteristics can be constructed. In particular, an asymmetric unimodal fuzzy number obtained with a Gaussian function with a bounded carrier can be written as follows:

$$\mu_{UG^m}(x) = \begin{cases} 0, x \leq b - a_1 \\ \exp\left[\frac{(b+a_1-x)(x-b+a_1)-a_1^2}{(b+a_1-x)(x-b+a_1)}\right], b - a_1 < x \leq b \\ \exp\left[\frac{(b+a_2-x)(x-b+a_2)-a_2^2}{(b+a_2-x)(x-b+a_2)}\right], b \leq x < b + a_2 \\ 0, x \geq b + a_2 \end{cases} \tag{13}$$

where $b$ is a modal value for membership function $\mu_{UG^m}(x)$ and $a_1, a_2$ are parameters which map the parts to the left and right of the fuzzy number carrier with membership function $\mu_{UG^m}(x)$, respectively.

In general, the results of this study suggest prospects for further development of the issues under study. Such a comprehensive approach can help to clarify and detail the results of analyses and decision making under conditions of uncertainty. In addition, the study can be used in other areas, for example, to solve optimization and planning problems under uncertainty and ambiguity. An important direction for further research may be to investigate the effectiveness of applying this approach in practice.

Let $U$ be a universal set (universum), $X$ be some subset of it. $(X \subseteq U)$, $R \subseteq X \times X$ is an equivalence relationship (in the framework of the theory in question it is interpreted as a relationship of indistinguishability), and $[X]_R$ is element equivalence class $X$ (in the theory of coarse sets, each equivalence class generated by a relation, $R$, is called an elementary set). Then, each set according to the described approach can be represented by two sets: $R$, lower $(\underline{R}X)$ and $R$, top $(\overline{R}X)$, with the following approximations:

$$\underline{R}X = \{x | [x]_R \subseteq X\}, \tag{14}$$

$$\overline{R}X = \{x | [x]_R \cap X \neq 0\}, \tag{15}$$

where

$$\underline{R}X \subseteq X \subseteq \overline{R}X. \tag{16}$$

Thus, $R$ is the lower approximation which consists of elements that definitely (exactly) belong to set $X$, whereas up to $R$, the upper approximation includes elements that possibly belong to set $X$.

Based on a given pair of approximations, $U$, the universe, can be divided into three subsets:

$POS_R(X) = \underline{R}X$: $R$ is the positive zone, the elements of which correspond to the definition, and $R$ is the lower approximation, which definitely (exactly) belongs to set $X$; $NEG_R(X) = U \backslash \overline{R}X$: $R$ is a negative zone, the elements of which certainly (exactly) do not belong to set $X$; and $BN_R(X) = \overline{R}X \backslash \underline{R}X$: $R$ is the boundary zone whose elements belong to the upper $R$ approximation and do not belong to the lower $R$ approximation.

The analysis of the logical scheme, on which the concept of a rough set is based, demonstrates the possibility of using it for modeling the initial financial and economic parameters of real investments in situations with a high degree of uncertainty and risk. The coarse set describes a set of possible situations and the expected results of investments, considering possible uncertainties.

In this context, rough sets can be used to identify factors that may affect the resulting financial performance of an investment. For example, rough sets can be used to identify possible risks and uncertainties that may affect investment outcomes, such as changes in the economic environment, changes in market trends, etc.

Thus, the concept of a rough set can be useful for determining appropriate investment and risk strategies in highly uncertain environments, as well as for managing financial risks in a given situation.

Let $X$ be some initial financial and economic parameter of the investment project, the value of which should be estimated. Let us further assume that the estimate of this parameter is a fuzzy number, denoted as $\widetilde{X}$. According to our proposed approach, the desired fuzzy estimate can be found utilizing the following steps.

The expert analyzes the raw data and determines the parameters that are close to the real values. The distribution of values is then plotted, and the maximum and minimum values within the probability density function are determined. This interval is used as the internal reference. To determine the interval more accurately, statistical methods such as confidence interval estimation can be used.

An initial value is set, which corresponds to the above condition. It is thus assigned the status of the base (base). New values are then added to the base, to the left and the right. The process continues until no difficulties are encountered when trying to advance further. The internal reference interval implies an interpretation as an external approximation of the kernel and an internal approximation of the carrier for the fuzzy estimate being sought.

Thus

$$\underline{x}^* \leq \underline{x}^1 \leq \overline{x}^1 \leq \overline{x}^*, \tag{17}$$

$$\underline{x}^0 \leq \underline{x}^* \leq \overline{x}^* \leq \overline{x}^0, \tag{18}$$

where $[\underline{x}^*, \overline{x}^*] = \overline{\overline{X}}^*$ is the internal reference interval for fuzzy parameter estimation $X$; $\underline{x}^*, \overline{x}^*$ are the lower and upper limits of the internal reference interval, respectively $\overline{\overline{X}}^*$; $\left[\underline{x}^1, \overline{x}^1\right] = \overline{\overline{X}}^1$ is fuzzy number kernel $\widetilde{X}$; $\underline{x}^1, \overline{x}^1$ are the lower and upper limits of the interval, respectively, embodying the kernel of fuzzy number $\widetilde{X}$; $\left[\underline{x}^0, \overline{x}^0\right] = \overline{\overline{X}}^0$ is the closure of fuzzy number carrier $\widetilde{X}$; and $\underline{x}^0, \overline{x}^0$ are the lower and upper limits of the interval, respectively, which realizes the closure of fuzzy number carrier $\widetilde{X}$.

1.  To the left and right of the interval obtained in the previous step, the semi-infinite intervals of values of the analyzed financial and economic parameters are determined, which, purely in principle, cannot be realized. These are referred to as the lower and upper non-opportunity intervals, respectively. For example, in the case of the lower non-reference interval, it can be performed as follows. To the left of the inner reference interval, an initial value is set which corresponds to the above condition. This value is assigned a base (base) status. New values are gradually and sequentially added to the right of the base. The process continues until there are no difficulties encountered in trying to proceed. The upper non-base interval can be constructed similarly to how the lower non-base interval is.

A formalized description of these theoretical constructs can be derived as follows: $\overline{\overline{X}}^{(-)} = \left((-\infty, \overline{x}^{(-)}\right], \overline{\overline{X}}^{(+)} = \left[\underline{x}^{(+)}, +\infty\right)$ are the lower non-supported interval and the upper non-supported interval with respect to the fuzzy parameter estimate $X$, respectively; $\overline{x}^{(-)}$ is the upper limit of the lower unsupported interval; and $\overline{x}^{(+)}$ is the lower limit of the upper non-support interval.

2. The interval of the analyzed financial and economic parameters is determined, which concerns the internal reference interval; then, the values which presented difficulties when assigning them to internal reference intervals and non-reference intervals, as well as the boundaries of the latter: $\overline{x}^{(-)}$ and $\underline{x}^{(+)}$, are determined. This interval is called the outer reference interval. The formula for finding the outer reference interval can be written as follows:

$$\overline{\overline{X}}^{**} = [\underline{x}^{**}, \overline{x}^{**}] = \left\{ \overline{x}^{(-)} \right\} \cup \left( \overline{x}^{(-)}, \underline{x}^{*} \right) \cup [\underline{x}^{*}, \overline{x}^{*}] \cup \left( \overline{x}^{*}, \underline{x}^{(+)} \right) \cup \left\{ \underline{x}^{(+)} \right\} = \left[ \overline{x}^{(-)}, \underline{x}^{(+)} \right], \tag{19}$$

where $\overline{\overline{X}}^{**} = [\underline{x}^{**}, \overline{x}^{**}]$ is the external reference interval for fuzzy parameter estimation $X$; $\underline{x}^{**}, \overline{x}^{**}$ are the lower and upper limits of the outer reference interval, $\overline{\overline{X}}^{**}$, respectively, on the basis of (19): $\underline{x}^{**} = \overline{x}^{(-)}, \overline{x}^{**} = \underline{x}^{(+)}$. The outer reference interval implies an interpretation as an outer approximation of the carrier for the fuzzy estimate sought; this is

$$\underline{x}^{**} \leq \underline{x}^0 \leq \overline{x}^0 \leq \overline{x}^{**}. \tag{20}$$

3. Thus, the inner reference interval can be considered to coincide with the kernel, and the outer reference interval can be considered to coincide with closure of the carrier of the desired fuzzy estimation. It is logical to refer to the presented approach of finding fuzzy estimates of initial financial and economic parameters of real investments as the method of reference intervals.

This method is based on the use of fuzzy numbers and their intervals, which reflect uncertainty and ambiguity in initial data. The reference intervals are determined on the basis of expert estimates or statistical data, and may include parameters such as investment cost, profitability, payback period, etc.

Furthermore, based on the reference intervals, an analysis of risks and determination of the probability of achieving the desired level of return was carried out. This can enable investors to make more informed decisions and reduce the risk of losing invested funds.

Thus, the method of reference intervals is an effective tool for assessing investment opportunities and enabling investors to make decisions based on fuzzy and uncertain data.

## 4. Results

Thus, the following results were obtained as part of this study:

1. The authors have developed a theoretical framework and a practical proposal for the production of radically new products.
2. The authors found that the sustainable development of country leaders, such as China, the USA, India, and Japan, occurs due to the fact that they export about 50% of the total output of knowledge-intensive products, which dominates the markets constantly. The organizations of these countries are constantly updating these products, increasing their competitive advantages.
3. The basic approaches and economic toolkit have been developed, aiming to ensure the stable development of the organization when applied in practice.
4. The formation of a technological platform for the industry is proposed, the creation of which represents an opportunity to create radically new products through the application of radical competencies that can form new markets or dominate existing ones. The technological platform of the industry is a set of inter-related processes creating value for the consumer in various sectors of the economy, based on radical competencies and key technologies of the product life cycle and the basic principles of cooperation between participants.
5. A toolkit was developed based on a mathematical model to manage the effectiveness of investment in the creation of radical new products, ensuring the formation of organizations working towards progressive, forward-looking, and stable economic development. A conceptual mathematical model for assessing the criteria of economic efficiency of the project was proposed as a toolkit.

The results of the present study, when applied in practice, could lead to the formation and the effective management of an organization with technological superiority, promoting stable economic development in modern economic conditions.

## 5. Discussion

First, according to the results of the research work of scientists, we consider the issues of directions of improvement of their work in order to meet today's requirements of relevance and take into account the factors and risks that are associated with the creation of sustainable economic development.

Second, it should be noted that it is necessary to analyze the relationship of sustainable economic development of leading countries with the production of knowledge-intensive high-tech products, their renewal, and the creation of competitive advantages.

Third, an important point of discussion is the issues related to the creation and production of fundamentally new products in terms of their characteristics and parameters that determine the needs and their dominance in the markets.

Fourth, it is necessary to consider the question under what economic processes occurring in the organization lead this organization to its advanced development. By advanced development of organization, we understand the process that includes the creation of fundamentally new products, organization of its production on the basis of innovative technical and technological solutions, physical principles with highly competitive advantages, and consumer properties that ensure the creation of new markets.

The processes associated with the creation of fundamentally new products and advanced development of the organization are closely connected with their resource support, which should be aimed at the creation of fundamentally new products.

Thus, there is a need to consider the issues associated with the formation of mathematical apparatus, which will allow the effective management of investment in the creation of fundamentally new products for sustainable economic development.

## 6. Conclusions

To demonstrate the implementation of the method under consideration, an example situation is presented in which the specialists of an enterprise were tasked with analyzing the economic feasibility of an investment project of the production and sale of a radically new product. Due to the lack of relevant statistical data and the dynamism of market conditions, most initial financial and economic parameters of the project involve estimates in fuzzy forms. In this case, the price of the product is predicted on the basis of the method of reference intervals. After the necessary analytical procedures were performed for the specified parameters, the following estimates of reference intervals were obtained: for the internal reference interval, $\overline{\overline{P}}^* = \left[ \underline{p}^*, \overline{p}^* \right] : \underline{p}^* = 700$ c.u., $\overline{p}^* = 900$ c.u.; for the outer reference interval, $\overline{\overline{P}}^{**} = \left[ \underline{p}^{**}, \overline{p}^{**} \right] : \underline{p}^{**} = 700$ c.u., $\overline{p}^{**} = 1000$ c.u.

In addition to the choice of a particular variant of the fuzzy number for unambiguous reproduction of the desired fuzzy score, in each case, the hypothesis that the inner reference interval coincided with its kernel and the outer reference interval coincided with the closure of its carrier was accepted.

The resulting generalized economic–mathematical model of the economic evaluation of the volume of project investment, aimed at creating radically new products and ensuring the advanced stable development of an organization, represents the simultaneous consideration of discrete and continuous risk factors, where the influence of the former is taken into account using a scenario methodology, and the latter, depending on the model setting, is based on probability theory, interval analysis, or the theory of fuzzy sets.

The application of this model on the basis of the approaches developed above and tools in practice will improve the management processes that ensure sustainable economic development of the organization.

**Author Contributions:** Conceptualization, Z.C. and A.C.; methodology, Z.C., A.O., A.B. and A.C.; formal analysis, A.C. and Z.C.; resources, Z.C. and A.C.; data curation, Z.C., A.B. and A.O.; writing—original draft preparation, Z.C.; writing—review and editing, Z.C., A.B. and A.O.; project administration, Z.C. All authors have read and agreed to the published version of the manuscript.

**Funding:** This paper has been supported by the RUDN University Strategic Academic Leadership Program.

**Institutional Review Board Statement:** Not applicable.

**Informed Consent Statement:** Not applicable.

**Data Availability Statement:** For the first time, it has been established that the sustainable development of organizations is provided by technological superiority based on the creation and production of radically new products that can form or dominate new and existing markets.

**Acknowledgments:** This study was supported by the RUDN University Strategic Academic Leadership Program.

**Conflicts of Interest:** The authors declare no conflict of interest. The funders had no role in the design of the study; in the collection, analyses, or interpretation of data; in the writing of the manuscript, or in the decision to publish the results.

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
