# Peer review of "The Creation of Fundamentally New Products as a Factor of Organizations’ Sustainable Economic Development"

_sustainability, doi:10.3390/su15129747_

Round 1

Reviewer 1 Report

The literature section should be developed and supported by current publications.. The fuzzy logic technique used should be explained in more detail. conclusions and recommendations section should be developed. also resources should be updated with more recent publications in this area.

Minor errors related to the language of the article should be corrected

Author Response

We apologize for any lack of clarity in our previous version that may have caused misunderstanding. However, at present there are very few such publications within the scope of the topic presented. We are the first researchers in this field. Unfortunately, there are currently no more relevant manuscripts on this topic. 

Reviewer 2 Report

The authors are faced with an interesting problem nowadays, namely the development of tools for the effective use of investments in creating radically new products to ensure the formation of organizations of advanced, sustainable economic development.

Although the paper is interesting, from my point of view, there are some directions for improvement.

1. The introduction is correctly done.

2. The second chapter, Literature Review, and Hypothesis Development, is well written.

3. The Research Methodology is still hard to follow, the equations do not include the meaning of the terms. A clarification of the proposed models would be helpful to the reader – also an explanation of the terms.

4. The Results Case Studies and Discussions chapter is, in my view, underrated. The detailed elaboration of the novelty of the study, but also the limits, as well as a detailed presentation of the conclusions would contribute to increasing the quality of the paper.

Author Response

Thank you for your positive feedback on our manuscript. We are glad that our efforts have been recognized. Your feedback is greatly appreciated and will help us to continue improving our manuscript. Explanations of terms added.

Reviewer 3 Report

The clarity is missing in the paper.

Continuation in the paper should be improved.

Plagiarism is high and it has to be reduced.

English language usage is fine.

Author Response

We agree. In the introduction we added the goal of the work for a clearer understanding.
The purpose of this study is to determine the theoretical and practical foundations of the creation of radically new products and the implementation of the processes of advanced development of enterprises, taking into account the application of mathematical modeling of resource provision.

Reviewer 4 Report

Line 167 mention Hypothesis. Please state hypothesis clearly and refer to that in the conclusion.

How are you utilizing the 3 groups in line 244? Why 3?

I miss a link back to introduction with GDP in chapter 4.

Looks good

Author Response

Thank you for pointing out those issues. We apologize for any confusion caused.  We formulated the hypothesis more precisely and we referred to it in manuscript. We distinguish exactly three groups of methods for obtaining subjective probabilities. The first group consists of methods in which the expert, in answering the questions posed, directly operates with the probabilities of the events being analyzed. It is the most extensive in its composition. The methods of the second group consist in obtaining event probabilities based on experts' decisions in some hypothetical choice situation, in which the result of the alter-natives under study is a realization of a random variable. The third group includes methods that combine aspects of the previous two ap-proaches, i.e., the expert has to operate with both probabilities and utilities.

Reviewer 5 Report

The article presented for review requires, in my opinion, a two-way evaluation. Firstly, the evaluation should concern the mathematical layer, in which some mathematical model is presented.  Secondly, the evaluation should concern the economic-application layer.

In my review, I will focus mainly on the latter.

In my review, I would like to focus on the weaknesses that I noticed while reading this article:

1. The title of the article - it is not clear what the authors mean by "sustainable economic development of organisations". The vast majority of the literature review, para. 2 of the article, is about linking the concept of "sustainable development" to the macro- or meso-economic level. Hence, in my opinion, the literature review needs to be supplemented/expanded accordingly. Furthermore, the title of the article itself should be rethought.

2. Abstract + Introduction - in these sections, the authors do not clearly formulate the purpose of the article. This should be supplemented.

3. Introduction - especially in this part, attention should be paid to the linguistic and stylistic correctness of the formulated sentences. There are many linguistic errors in the text resulting from incorrect sentence constructions. The article needs a thorough linguistic revision.

Literature review - the way sentences are constructed in this part of the article is unacceptable.  The vast majority of paragraphs begin with a parenthesis. All such sentences should necessarily be rephrased. One could start, for example, with the names of the cited authors. Furthermore, it is not necessary to create an independent paragraph for each item cited. The literature review serves the purpose of critically evaluating the scientific output to date, not enumerating a few items thematically related to the article. As I have already written in para. 1, the use of such literature in the article is questionable. It is not directly linked to the title of the article. Or the authors were not able to convincingly demonstrate that, from the point of view of the article's topic, the cited literature is useful.

5. Table 1 - the columns with GDP (PPP) values are unreadable, the table should be rebuilt. The last column can be dropped, the values presented there are a duplication of the values in the earlier column.

6 The hypothesis is not correctly formulated. The sentence on page 4 is not a research hypothesis.

7. Research methodology - too little space in the article is devoted to the factors - K1 to K4. The mathematical proof is described in great depth. However, in my opinion, the economist would like clear guidance on what follows from the application of this model. What is missing in the article is the reference of the mathematical model to real world conditions. This significantly limits the citability and interest of the article.

In summary, the economic and application layers of the article are very weak - poor literature review, not directly linked to the title of the article. No application of the mathematical model.

The mathematical layer is an asset, but this alone cannot account for the value of the article. Especially as the abstract and title suggest the economic direction of the scientific argument.

The article needs thorough improvement and re-review.

The article needs a thorough linguistic revision.

Author Response

Thank you for pointing out those issues. We apologize for any confusion caused. In this revised version, we have made comprehensive revisions and improvements in response to the issues you raised, and carefully checked for any other possible errors in the manuscript. We rethought the very title of the manuscript.

Under sustainable economic development of the organization we understand the organization, which is able to carry out work related to the renewal of products, the creation of radically new products at the expense of its own resources. The implementation of radically new products in the market in real time allows to increase the income of the enterprise, as well as provides in the medium and long term increase in the profitability of the enterprise. 

We have deliberately not disclosed in detail the economic nature of K1-K4 in this article due to the fact that these factors are described in detail in the book "New Economy of the Product Life Cycle : Innovation and Design in the Digital Era" Springer International Publishing A&G (2020). Authors: Andrey Tyulin and Alexander Chursin.

And we would like to inform you that we have carefully reviewed and revised the manuscript, paying close attention to clarity of language, fluency, and readability, including grammar, spelling, punctuation, and structure of speech. Thank you very much for giving us the opportunity to correct our mistake.

Round 2

Reviewer 2 Report

In general, the authors followed the reviewers' comments.

To improve the paper, the Conclusion Section should be detailed.

Author Response

Thank you very much for your comments. We have reviewed our manuscript and improved it. We have made the conclusion more detailed.

Reviewer 5 Report

On the language side, there are still some shortcomings in the article. The mixing of British English with American English indicates that the article has not undergone professional linguistic proofreading. There are some typos.

The results in the article are still not referenced to the work of other authors. The real Discussion section of the article is missing. This part of the article should be expanded.

On the language side, there are still some shortcomings in the article. The mixing of British English with American English indicates that the article has not undergone professional linguistic proofreading. There are some typos.

Author Response

Thank you very much for your comments. We have revised our manuscript and improved it. We added a Discussion section and we expanded the manuscript.  

We apologize, but we used the English version of the MDPI. MDPI Author Services has edited our text into English.